# An Item Response Theory Analysis of the Wisconsin Card Sorting Test in Normal Aging, Alzheimer’s Disease and Parkinson’s Disease: Neurophysiological Approach

**DOI:** 10.3390/jpm12040539

**Published:** 2022-03-29

**Authors:** Juan Luis Sánchez-Rodríguez, Raúl Juárez-Vela, Iván Santolalla-Arnedo, Regina Ruiz de Viñaspre-Hernandez, Juan Luis Sánchez-González

**Affiliations:** 1Department of Basic Psychology, Psychobiology and Methodology, Faculty of Psychology, University of Salamanca, 37005 Salamanca, Spain; jlsanch@usal.es; 2Department of Nursing, Research Group in Care, GRUPAC, 26004 Logroño, La Rioja, Spain; reruizde@unirioja.es; 3Department of Nursing and Physiotherapy, Faculty of Nursing and Physiotherapy, 37007 Salamanca, Spain; juanluissanchez@usal.es

**Keywords:** psychometrics, neuropsychology, validation study

## Abstract

The Wisconsin Card Sorting Test (WCST) is widely used to assess executive function, specifically mental flexibility. Item Response Theory (IRT) has several advantages over classical test theory, including the invariance of the measure. This study aimed to apply IRT to study the psychometric properties of WCST in control subjects and patients with idiopathic Parkinson’s disease (PD) and Alzheimer’s disease (AD). The sample consisted of 86 controls, 77 Parkinson’s disease subjects, and 155 AD subjects. The Rasch model was applied to binary data from the conversion to percentiles adjusted for age and schooling. The R program was used to calibrate the difficulty parameter. The characteristic curves of the items and the estimation of the difficulty parameter for each diagnostic group were obtained. WCST makes it possible to separate between different skill levels among the three diagnostic entities and its application may be useful for other neuropsychological tests. In conclusion, WCST has good clinical sensitivity and excellent discriminant validity in the groups under study, making it possible to separate them between different levels of ability or latent trait. Its application may be useful for other neuropsychological tests.

## 1. Introduction

The term executive function is frequently used to refer to a series of higher cognitive activities carried out by a complex and polyvalent mental system, with a neurological basis in the prefrontal cortex, especially the cingulum and anterior neocortical areas [1]. It is a system that deals with organizing the operation of ‘controlled’ processes with information of any modality [2,3,4]. A wide variety of skills has been included within the so-called executive functions, such as the ability to set goals, develop action plans, flexibility of thought, inhibition of automatic responses, self-regulation of behavior, and verbal fluency [5,6,7]. As such, the deficits of these functions can limit the individual’s ability to maintain an independent and productive life, even if other cognitive abilities are intact [8].

Currently, changes in cognitive functioning caused by aging are increasingly prevalent [9], especially those that occur in the most complex cognitive functions. These changes are manifested in broad characteristics: a slowdown in information processing, a decrease in attentional capacity, a decline in some aspects of memory, and in so-called executive functions. However, it is necessary to clarify that this deterioration is not global or generalized since there are functions that slowly progressively decline throughout adult life, others are maintained until very late stages, and, finally, there are some that not only do not deteriorate but even improve over time [10,11]. Parkinson’s Disease (PD), in addition to involving motor deficits from the onset of the disease, also affects different specific cognitive processes such as visuospatial and visuoperceptive areas, executive functions, and certain aspects of memory [12]. Deficits in executive functions are observed from the early stages of the disease, especially with regard to the manipulation and monitoring of information, whether verbally or visually. In general, patients with Parkinson’s disease produce fewer correct categories and a higher number of perseverative errors in the Wisconsin Card Sorting Test (WCST), as well as difficulties in organizing, managing, and substituting concepts for other newer concepts [13]. Difficulties in inhibiting interference and a decrease in sustaining attention selectively have also been found [14]. On the other hand, executive functions are altered early in Alzheimer’s disease (AD), such that the patient has an inability to plan and execute goal-directed actions, as well as a significant lack of cognitive flexibility [15]. This type of deficit in frontal functions is one that causes the greatest impact on the functionality of the individual and on his or her ability to remain independent [16]. The capacity for abstraction is also affected, making it difficult for the patient to understand metaphorical or figurative language. The lack of inhibition of motor and verbal responses is also common in advanced stages, which can result in socially inappropriate behaviors, such as sexual disinhibition or coarse language [17,18].

Numerous instruments for measuring executive functions are available in the literature, including WCST. The main objective of this test is to assess executive function, especially mental flexibility. It was developed by Berg and Grant [19,20] to assess the ability of an individual to abstract when changing cognitive strategies in response to changing environmental contingencies. As for psychometric properties, it has been verified that the WCST presents discreet test–retest reliability since a practical effect is observed in five test scores [21]. In addition, it presents excellent inter-rater reliability in a sample of adult psychiatric patients [22]. From a psychometric perspective, all the works cited have applied the Classical Test Theory (CTT) to study the psychometric properties of WCST. This theory, despite being the best known and used in neuropsychology, is not without limitations [23]. One of the main limitations is that the measurements are not invariant with respect to the instrument used. The other main limitation within the classical framework is the lack of invariance of the properties of the tests with respect to the persons used to estimate them, so that the important psychometric properties of tests, such as item difficulty or test reliability, are a function of the type of people who is administering the test [24]. To overcome these limitations, the Item Response Theory model proposed by Rasch in 1960 was developed [25]. The key supposition in IRT models is that there is a functional relationship between the values of the variable measured by the items and the probability of obtaining the correct value, calling this function the Item Characteristic Curve (ICC) [26]. The values of the measured variable, whatever it is, are between minus infinity and plus infinity [24], whereas, in classical theory the values depended on the scale of each test, ranging from the minimum value obtainable in the test to the maximum.

We aimed to apply Item Response Theory in the Wisconsin Card Sorting Test in a sample of healthy controls and Parkinson’s disease and Alzheimer’s disease patients to assess the test’s psychometric properties.

## 2. Materials and Methods

### 2.1. Study Design and Ethics

We performed a cross-sectional study in order to identify the clinical sensitivity of WCST. The sample used in this study consisted of 318 subjects. They were divided into three groups: 86 healthy participants, 77 patients diagnosed with PD, and 155 patients diagnosed with AD. The following exclusion criteria were considered: (a) history of neurological disorders (except AD and Parkinson’s disease in the clinical groups); (b) uncorrected visual and hearing impairments; (c) history of alcohol or other drug abuse or dependence; and (d) a mental state score of less than 28 in healthy controls in the Mini Mental State Examination (MMSE) [27]. Information on relatives was obtained through a semi-structured interview. The participants with PD and AD were selected at the Neurology Service of the University Hospital of Salamanca (Spain). The study protocol was approved by the Institutional Review Board of the University of Salamanca and the study adhered to the Declaration of Helsinki.

### 2.2. Study Population and Clinical Characteristics

The healthy controls were selected from among the individuals accompanying the patients on their visits to this service who fulfilled the necessary inclusion criteria. All participants signed an informed consent form before being tested and they received no financial reimbursement or any other compensation. Eighty-six healthy elderly controls (age range: 50–88 years) were studied, who at the time of examination were carefully screened to exclude those with dementia or any other medical conditions that would affect their neuropsychological performance. Seventy-seven patients were diagnosed with Parkinson’s disease following the clinical criteria for Parkinson’s disease [28]; that is, they responded to the presence of clinical symptoms: resting tremor, bradykinesia, rigidity, and postural instability. We excluded subjects who had parkinsonian extrapyramidal disorder, accompanied by characteristic signs of involvement of other areas of the central nervous system or an unusual evolution of the disease. Disease severity was assessed with the Hoehn and Yahr scale [29], with results of 2.16 ± 0.76. In connection with levodopa treatment, at the time of the study, all patients were taking a dose of between 250 and 1000 mg/day.

One hundred and fifty-five patients with a diagnosis of probable AD fulfilled the DSM V criteria (Diagnostic and Statistical Manual for Mental Disorders] [30] and NINCDS-ADRDA criteria (National Institute of Neurological and Communicative Disorders and Stroke-Alzheimer’s Disease and Related Disorders Association). We decided to consider both the groups of criteria for the diagnosis (DSM-V) and classification (NINCDS-ADRDA) of AD, since in addition to being the criteria most used internationally for the clinical diagnosis of dementia, they show high sensitivity (81%) and an acceptable level of specificity (70%) for AD [31].

The majority (99%) of study participants were right-handed, as confirmed by the Edinburgh Handedness Inventory [32]. The ethnic background of all participants was Caucasian, and all were living in Spain. Methodologically, the study was carried out in several stages. In the first stage, patients who presented at the Neurology Service and expressed subjective complaints about memory were evaluated by neurological examination and neuropsychological assessment. The patients were evaluated again after 6 months, enabling us to include patients in the category of probable AD. However, only the data obtained in the first evaluation were used in this research study. For the participants with AD, diagnostic tests were used in order to exclude those individuals with possible secondary dementia. Those included in the tests were as follows: thyroid function, luetic serology, levels of vitamin B12, and folic acid. All participants with AD were examined neuroradiologically by computed tomographic scans. According to the neuroradiological reports, all participants included in the clinical sample showed structural lesions consisting of the presence of different degrees of cerebral atrophy, which are mainly cortical.

### 2.3. Instruments

The MMSE was used to select the study participants, a measure of global cognition in its adapted Spanish version [27]. The study cut-off, adjusted for age and education, was 28 in cognitively healthy adults.

WCST was administered and scored by neuropsychologists specifically trained for this project, following the original WCST manual [33]. This test is considered a measure of executive functions and cognitive flexibility, as it requires 4 stimulus cards and 128 response cards containing figures of various shapes (circle, cross, triangle, and star), colors (green, blue, red, and yellow), and different numbers of shapes (one, two, three, and four). The subject must match the response cards with the stimulus cards, deducing the criteria to do so correctly from the information provided by the examiner in each trial. In this test, multiple performance scores were obtained and linked to different cognitive processes: the number of categories (inability to maintain an adequate strategy); the percentage of persevering responses (persistence in responding according to an incorrect criterion); the percentage of persistent and non-persistent errors; and the percentage of responses at the conceptual level (conceptual efficacy). There is no time limit on administering the test.

### 2.4. Data Analysis

The Statistical Package for Social Sciences (SPSS v.25, New York, NY, USA) was used to investigate group differences in demographic and MMSE scores. An analysis of variance (ANOVA) with post hoc analysis (Bonferroni) was used to compare sociodemographic and neuropsychological data among groups. Gender differences were assessed using chi-square tests. The significance level was set at <0.001. The Item Response Theory model applied in this study was the One Parameter Logistic Model, which is also known as the Rasch Model (1960). It is derived from the prediction of the probability of responding correctly to an item from the difference in the attribute between the level of the person (*θ*) and the level of difficulty of the item (parameter b). That is, the probability of obtaining a correct item depends only on the level of difficulty of the said item and the level of the subject in the measured variable. Item difficulty indicates how much of a trait is required to solve the item successfully. It describes where the item is situated in the ability scale. The Rasch model is defined by the following equation:$$p_{i}\left(\theta \right)=\frac{e^{D\left(\theta -b_{i}\right)}}{1+e^{D\left(\theta -b_{i}\right)}}$$
where*P_i_*(*θ*) is probability of hitting an item for a value of *θ*;*B_i_* is difficulty index of item _*i*_.

The model’s estimation of the parameter makes it possible to evaluate the technical quality of the test. The difficulty index was estimated separately for each diagnostic group (controls, Parkinson’s disease, and Alzheimer’s disease). This parameter defines ICC, where each point of the curve represents the probability of hitting the item with a certain skill level or latent trait of the subject. ICC characterizes the specific properties of each item. In this study, ICCs were defined for each of the WCST variables.

## 3. Results

Demographic data and MMSE scores are provided in Table 1. Participants in the control group were significantly younger than Parkinson’s disease and AD patients. The slight difference between groups in terms of gender and years of education was nonsignificant. Patients’ MMSE score were, as assumed from diagnosis of Parkinson’s disease and AD, lower than those for the control group.

Regarding the results from applying IRT, a good fit of the model was observed in the three groups studied, in the case of the controls (−268.5 Log likelihoods, Akaike information criterion (AIC) 559.01, and Bayesian information criterion (BIC) 586.01), in the subjects with Parkinson’s disease (−444.30 Log likelihoods, AIC 908.61, and BIC 932.04), and, finally, in the AD group (−879.30 Log likelihoods, AIC 1778.60, and BIC 1809.03). The parameters estimated using the Rasch Model for each WCST score are presented in Table 2. Regarding the difficulty of the items, the most difficult items are located in the group of patients with Parkinson’s disease for most of the variables studied, except in the case of perseverative responses and perseverative errors, with higher values in this parameter, while the easiest ones are found in the control subjects. In the group of control subjects, discrimination was 3.51 (Std error 0.48), in the subjects with AD, it was 1.98 (Std error 0.18), and in the subjects with PD, it was 1.78 (Std error 0.21).

Figure 1 shows the ICCs of all variables studied by diagnostic groups. The ICCs show a similar pattern between Parkinson´s disease and Alzheimer’s Disease, although they differ in the position of the curve, which is more to the right side in the ability scale for the Parkinson’s disease group; therefore, it more difficult for these subjects in most of the variables. Control subjects acquire the correct response quickly and with low levels of the latent trait.

## 4. Discussion

WCST is widely used to measure executive functions, particularly those that are related with the search for problem-solving strategies by adapting to changing external situations [34]. To date, all published studies on the psychometric properties of this test are based on the application of CTT. In the present study, we use IRT to study WCST in three diagnostic groups: normal aging, Parkinson’s disease, and Alzheimer’s disease. In IRT, the estimation of the model parameters evaluates the technical quality of the items separately. In this case, when applying the One Parameter Logistic Model or the Rasch Model, the difficulty index determines how well the items work [35]. The control subjects show a lower difficulty index in relation to the subjects with Parkinson’s disease or AD. This finding indicates that, with a low skill level, a correct answer is achieved. Likewise, IRT studies the subject’s ability, which is also known as a latent trait. The best representation of this parameter is found in the characteristic curves of the items, which reflect the degree of competence of the subjects in WCST. In addition, this level of ability is free from the effect of age and schooling, since we worked with normative data adjusted for age and schooling, which were subsequently dichotomized into correct–incorrect. Dichotomous or binary data are often used in neuropsychological clinical practice [36], for example, in the presence–absence of impairment. The ICCs presented show different levels of ability for each group studied. The position of the curve is found at lower levels of ability in the controls, medium levels in the PD, and at high levels in the AD group, which are represented more to the right of the latent trait scale. That is, the control subjects can solve the test successfully, requiring little effort. In the case of subjects with Parkinson’s disease and AD, the probability of success between one pathology and another decreases, the probability being lower in the case of subjects with PD, where the normal level of a latent trait is required to reach an adequate response. The ICCs have different forms and positions. Therefore, from the IRT perspective, WSCT is sensitive in discriminating between different ability levels (discriminant validity), which include subjects with high levels of functioning and those with Alzheimer’s disease and Parkinson’s disease.

It is well documented that the performance on WCST tends to decline in older subjects, but the cause of this poor performance has not been well established. Several studies [37,38,39] carried out to determine these changes found that subjects from the age of 50 presented a worse performance in variables, including perseveration errors and perseverative responses. On the other hand, subjects older than 60 years completed fewer categories than subjects younger than 40.

Traditionally, CTT has been the most widely used psychometric approach in neuropsychology, due in part to its simplicity and popularity. Despite this, CTT has several limitations that have been attempted to be resolved with the appearance of IRT, which offers many advantages. The main one is the invariance of the measure in two aspects: both the invariance for the test and concerning the normative group [26]. Another advantage is that it allows us to estimate the precision with which each item and each test measure the different levels of ability [40], allowing us to have different error measures for each individual and/or skill level [41].

On the other hand, IRT is based on nonlinear models that are generally logistic [42]. Starting from the assumption that cognitive functioning in aging has a curvilinear nature and global cognitive impairment in AD and Parkinson’s disease follows a nonlinear course [43,44,45], the use of IRT brings us closer to this problem [46,47].

## 5. Conclusions

In conclusion, WCST has good clinical sensitivity and excellent discriminant validity in different groups of subjects: normal aging, Alzheimer’s disease, and Parkinson’s disease. In this sense, it makes it possible to separate the different levels of skill or latent traits. This study is, to our knowledge, the first to apply IRT to WCST, making an interesting contribution to the study of this test.

## 6. Limitations

We only focus on Parkinson’s and Alzheimer’s pathologies. Another range of neurological diseases may be taken into consideration. In addition, we have to take into account age as a confusion bias.

## Figures and Tables

**Figure 1 jpm-12-00539-f001:**
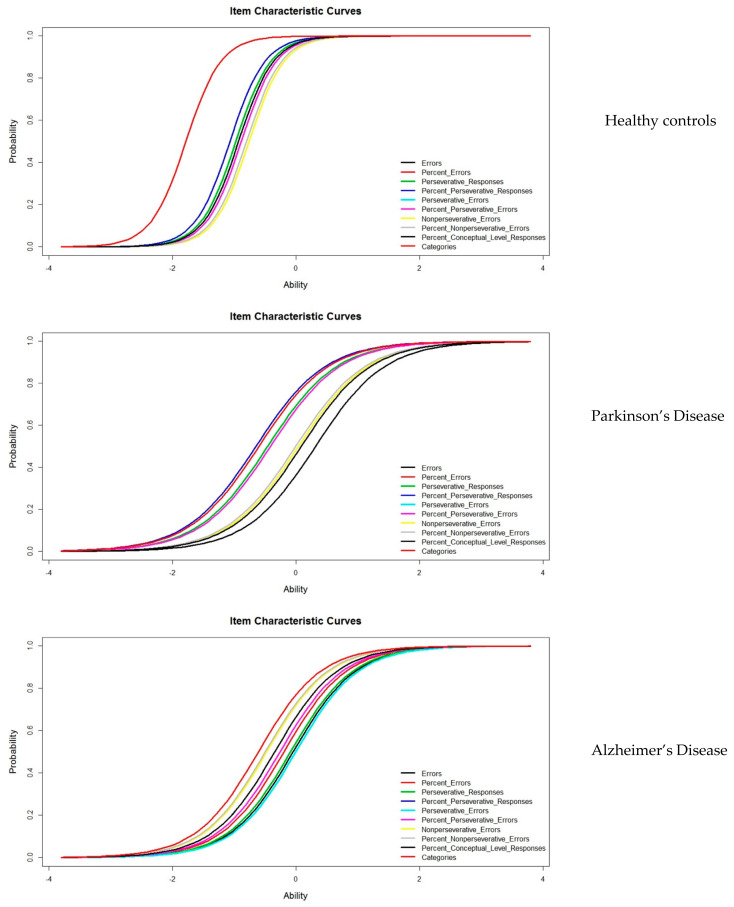
Item Characteristic Curves for the three groups studied.

**Table 1 jpm-12-00539-t001:** Demographic characteristics and MMSE score for the three groups studied.

	Control (*n* = 86)	PD (*n* = 77)	AD (*n* = 155)	Sign.
Age, years	70.71 ± 7.54	71.81 ± 7.26	74.75 ± 7.66	<0.0001
Female gender %	52	55	67	n.s.
Education, years	7.86 ± 3.506	7.31 ± 3.63	6.94 ± 3.19	n.s.
MMSE	33.45 ± 1.46	30.72 ± 4.29	26.16 ± 5.59	<0.0001

Data are presented in the following form: mean ± standard deviation; MMSE, Mini-Mental Status Examination; group differences were performed by analysis of variance, ANOVA, with post hoc analysis (Bonferroni); gender percentage comparison was performed by mean chi-square test with continuity correction; PD = Parkinson´s disease; AD = Alzheimer´s disease.

**Table 2 jpm-12-00539-t002:** Rasch Model parameter estimates for healthy controls, Parkinson’s disease and Alzheimer’s disease.

	Healthy Controls		Parkinson’s Disease		Alzheimer’s Disease	
	Difficulty	Std Error	Difficulty	Std Error	Difficulty	Std Error
Errors	−0.9733	0.1380	0.3179	1.5232	−0.0441	0.1353
% Errors	−1.0681	0.1435	−0.0044	−0.0215	−0.1934	0.1354
Perseverative Responses	−0.9732	0.1380	−0.4579	−2.2028	−0.0868	0.1352
% Perseverative Responses	−1.0681	0.1435	−0.6481	−3.0218	−0.3431	0.1368
Perseverative Errors	−0.8840	0.1336	−0.4116	−1.9914	−0.0014	0.1355
% Perseverative Errors	−0.8839	0.1336	−0.4116	−1.9914	−0.2570	0.1358
Nonperseverative Errors	−0.7583	0.1293	0.0409	0.2007	−0.4735	0.1390
% Nonperseverative Errors	−0.7992	0.1304	−0.0044	−0.0214	−0.4955	0.1395
% Conceptual Level Responses	−0.9279	0.1357	0.0863	0.4228	−0.3432	0.1368
Categories	−1.7806	0.2394	−0.5996	−2.8216	−0.6075	0.1424

## Data Availability

The authors confirm that the data supporting the findings of this study are available upon contact with the corresponding author.

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
