# Peer review of "An Item Response Theory Analysis of the Wisconsin Card Sorting Test in Normal Aging, Alzheimer’s Disease and Parkinson’s Disease: Neurophysiological Approach"

_jpm, 2022, doi:10.3390/jpm12040539_

Round 1

Reviewer 1 Report

Your study is interesting and significant. However, there are some comments to be considered. 

When writing an abbreviation in ms, write the abbreviation and the full name together when they first appear, and then use only the abbreviation. Please check the CTT, WCST, and IRT in ms.

In the discussion, you mentioned that there was only a test to which Classical Test Theory was applied in the previous study, so in this study, the WCST was tested by applying the Item Response Theory. It would be better if these reasons for this study were described in the introduction part. Please describe the limitations of applying CTT and the advantages of applying ICT.

Please add more explanation regarding PD and AD with supporting reference (Line 196-200).

The position of the curve is found at lower levels of 196 ability in the controls, medium levels in the PD, and at high levels in the AD group, represented more to the right of the 197 latent trait scale. That is, the control subjects can solve the test successfully, requiring little effort. In the case of subjects 198 with PD and AD, the probability of success between one pathology and another decrease, the probability being lower 199 in the case of subjects with PD, where a higher level of latent trait is required to reach an adequate response. The ICCs 200 have different forms and positions.

Line 50-51. Please check this sentences line 50-51. (others. Newer. ?)

Author Response

Your study is interesting and significant. However, there are some comments to be considered. 

When writing an abbreviation in ms, write the abbreviation and the full name together when they first appear, and then use only the abbreviation. Please check the CTT, WCST, and IRT in ms.

Thank you very much for your comment. We have addressed it.

In the discussion, you mentioned that there was only a test to which Classical Test Theory was applied in the previous study, so in this study, the WCST was tested by applying the Item Response Theory. It would be better if these reasons for this study were described in the introduction part. Please describe the limitations of applying CTT and the advantages of applying ICT.

Thank you very much for your input. More information on both models has been added in the introduction section.

Please add more explanation regarding PD and AD with supporting reference (Line 196-200): The position of the curve is found at lower levels of ability in the controls, medium levels in the PD, and at high levels in the AD group, represented more to the right of the latent trait scale. That is, the control subjects can solve the test successfully, requiring little effort. In the case of subjects with PD and AD, the probability of success between one pathology and another decrease, the probability being lower in the case of subjects with PD, where a higher level of latent trait is required to reach an adequate response. The ICCs have different forms and positions.

Thanks for the great output, however, this section is regarding our result, indeed the curve is the modeling of our results, but we addressed it, and clarify.

Line 50-51. Please check this sentences line 50-51. (others. Newer. ?)

We have addressed it. Thank you so much.

Reviewer 2 Report

Overall, the manuscript seems to be drafted with some superficiality. Several issues (e.g., typos; both formal and theoretical inaccuracies; etc.) are present and should be addressed before considering the paper for publication; similarly, major English revision is needed.

Abstract

Check the formatting for the conclusions: it does not follow the formatting characteristics of the Abstract and ‘In conclusion’ is repeated twice.

Introduction

Overall, the introduction does not adequately cover the relevant literature, being quite superficial. Just to mention the main issue, the paper is focused on applying Item Response Theory to the WCST and on highlighting its advantages over Classical Test Theory but none of the two theories is adequately presented in the Introduction, leaving the reader uncapable of understanding the aim of the paper. Furthermore, in the background part related to Parkinson and Alzheimer diseases many statements are not supported by any reference. Additionally, the specific authors’ hypotheses are not presented at all. To be acceptable, a major revision of the Introduction is needed.

Below, I report some specific comments, even though I am not listing all the typos and references missing (just some examples). As mentioned above, besides these specific comments, the Introduction section needs major revision.

  • I recommend better consistency throughout the text in the choice of whether using acronyms or not. For instance, the acronym EF is defined but never used throughout the test as the authors use always the extended form ‘executive functions’. Similarly, even though they define the acronym ‘WCST’, they don’t always use it. The same can be said for the acronym ‘IRT’.
  • Line 32: A reference is needed.
  • Lines 37-38: it is said “That is why the performance of these functions can limit the individual's ability to maintain an independent and productive life, even if other cognitive abilities are intact (7).” This sentence does not make sense – maybe the authors meant that executive deficits could limit individuals’ ability to maintain an independent life?
  • Line 40-41: it is said “Nowadays, the changes that aging causes in cognitive functioning are becoming increasingly important, especially those that occur in the most complex cognitive functions.” Why so?
  • Line 42: it is said: “These changes are manifested in great characteristics (…)”. The use of the terms ‘great characteristics’ is inappropriate for what the authors are referring to.
  • Line 51-52: it is not understandable what the authors mean by saying “(…) substituting concepts for others. newer.”
  • Line 52: the statement “Difficulty in inhibiting interference and a decrease in sustaining attention selectively have also been found” lacks any reference.
  • The background on Alzheimer disease (lines 53-59) needs references.
  • Item Response Theory and Classical Test Theory should be adequately presented.

Method

  • Line 112: The acronym ‘MMSE’ must be defined.

Results

  • The presentation of results could be improved. So far, they are too concise and difficult to follow.
  • line 168: what is ‘EA’?

Discussion

The Discussion section is better developed than the Introduction; still it might be enriched with more theoretical integration of the results within the current literature. The Limits should be better addressed. Look also for typos / inaccuracies (e.g., Line 216: what is “TCT”? Did you mean “CTT”? Line 217: again, “TRI” stands for “IRT”?).

Author Response

Overall, the manuscript seems to be drafted with some superficiality. Several issues (e.g., typos; both formal and theoretical inaccuracies; etc.) are present and should be addressed before considering the paper for publication; similarly, major English revision is needed.

Thanks so much for your input. We have addressed it. Please find enclose the certification of English review.

Abstract

Check the formatting for the conclusions: it does not follow the formatting characteristics of the Abstract and ‘In conclusion’ is repeated twice.

Thanks so much. We have addressed it.

Introduction

Overall, the introduction does not adequately cover the relevant literature, being quite superficial. Just to mention the main issue, the paper is focused on applying Item Response Theory to the WCST and on highlighting its advantages over Classical Test Theory but none of the two theories is adequately presented in the Introduction, leaving the reader uncapable of understanding the aim of the paper. Furthermore, in the background part related to Parkinson and Alzheimer diseases many statements are not supported by any reference. Additionally, the specific authors’ hypotheses are not presented at all. To be acceptable, a major revision of the Introduction is needed.

Thank you very much for all the consideration provided. The manuscript has been revised and changed based on your comments. In addition, the language has been revised by Studio Moretto Group based in Brescia, Italy.

Below, I report some specific comments, even though I am not listing all the typos and references missing (just some examples). As mentioned above, besides these specific comments, the Introduction section needs major revision. I recommend better consistency throughout the text in the choice of whether to use acronyms or not. For instance, the acronym EF is defined but never used throughout the test as the authors use always the extended form ‘executive functions’. Similarly, even though they define the acronym ‘WCST’, they don’t always use it. The same can be said for the acronym ‘IRT’.

Thanks so much. We have addressed it

Line 32: A reference is needed.

Thanks so much. We have addressed it. 

Lines 37-38: it is said “That is why the performance of these functions can limit the individual's ability to maintain an independent and productive life, even if other cognitive abilities are intact (7).” This sentence does not make sense – maybe the authors meant that executive deficits could limit individuals’ ability to maintain an independent life?

Indeed, it was a mistake in the phrase. We have addressed it.

Line 40-41: it is said “Nowadays, the changes that aging causes in cognitive functioning are becoming increasingly important, especially those that occur in the most complex cognitive functions.” Why so?

Reading the sentence again, we have expressed ourselves badly. We want to emphasize that currently due to the increase of life expectancy in the population, more and more people suffer changes associated with aging, precisely in cognitive functions such as executive functions. That is to say, there has been an increase in the prevalence

Line 42: it is said: “These changes are manifested in great characteristics (…)”. The use of the terms ‘great characteristics’ is inappropriate for what the authors are referring to.

Thanks so much it has been replaced by: "These changes are manifested in broad characteristics"

Line 51-52: it is not understandable what the authors mean by saying “(…) substituting concepts for others. newer.”

Thanks so much, we have addressed it. It has been replaced by: "or other newer concepts". 

Line 52: the statement “Difficulty in inhibiting interference and a decrease in sustaining attention selectively have also been found” lacks any reference.

Thanks so much. It has been addressed 

The background on Alzheimer disease (lines 53-59) needs references.

Thanks so much. It has been addressed 

Item Response Theory and Classical Test Theory should be adequately presented.

Thanks so much. It has been addressed 

Method

  • Line 112: The acronym ‘MMSE’ must be defined.

Thanks so much. It has been addressed . 

Results

The presentation of results could be improved. So far, they are too concise and difficult to follow.

Thanks so much. With the English review by an English professional editing service, it has been clarified.

line 168: what is ‘EA’?

This is a mistake. We refer to Alzheimer's disease (AD). So sorry. 

Discussion

The Discussion section is better developed than the Introduction; still it might be enriched with more theoretical integration of the results within the current literature. The Limits should be better addressed. Look also for typos / inaccuracies (e.g., Line 216: what is “TCT”? Did you mean “CTT”? Line 217: again, “TRI” stands for “IRT”?).

Thanks so much. It has been addressed 

Dear reviewer thanks so much for your wonderful review. We aprecciate so much in order to improve the paper. 

Round 2

Reviewer 2 Report

Overall, I acknowledge the manuscript has been improved. Still there are some aspect that need to be addressed.

Method

There’s a bit of confusion with the subtitles used in the method section. Under the “Study design and Ethics” sub-paragraph, the Authors are not presenting information regarding the study design but rather inclusion and exclusion criteria as well as information regarding how the clinical population was recruited, which are more suitable to be included within the presentation of the study sample. Similarly, the following sub-paragraph (i.e., “Study Population and Clinical Characteristics”), together with information regarding the populations, includes a depiction of the methodological steps along which the study was conducted, which does not properly fit a paragraph focused on the study population.

Results

Line 155-156. The authors stated “Participants in the control group were significantly younger than PD and AD patients. However, in post hoc analysis (Bonferroni) these patients only differ between AD patients.” I have two main concerns.

  1. First, when saying “in post hoc analysis (Bonferroni) these patients only differ between AD patients” do they mean that the participants of the control group only differ from AD patients? From what they said before, in the method section, the control group is composed by healthy controls, not other patients.
  2. Second, do the authors think that age might influence their results? If yes, how? Did you control for it? In case you didn’t, you should do it or at least present this among the limitations of your study. By constrast, if they don't think age has an influence, why not? Generally, this aspect should be better addressed. Additionally, considering that the authors assessed differences among groups on demographic characteristics and they presented such results, I am assuming they thought that such variables might be of interest for their study. That also considering what they say about age in the Discussion section (from line 211).

Discussion

Lines 202-206: “The ICCs presented show different levels of ability for each group studied. The position of the curve is found at lower levels of ability in the controls, medium levels in the PD, and at high levels in the AD group, represented more to the right of the latent trait scale. That is, the control subjects can solve the test successfully, requiring little effort. In the case of subjects with PD and AD, the probability of success between one pathology and another decreases, the probability being lower in the case of subjects with PD, where a higher level of latent trait is required to reach an adequate response.” I have to main concerns regarding this paragraph:

  1. Looking at the second part of the paragraph (line 206) as well as at Figure 1 and at its description (from line 173), one can understand that PD patients are the ones who show the worse performance. If this is the case, why the authors states “at lower levels of ability in the controls, medium levels in the PD, and at high levels in the AD group, represented more to the right of the latent trait scale” (line 203-204)?
  2. Why lower levels of ability correspond to better performance? Can the authors better explain this?

Line 208. The Authors states “the WSCT is sensitive in discriminating between different ability levels (discriminant validity). That is, subjects with high levels of functioning, with Alzheimer's disease and Parkinson's disease.” When mentioning “subject with high levels of functioning” they are actually referring to (old) healthy controls. Thus, it is inappropriate to talk about high levels of functioning; rather they should talk about normal functioning at the most. 

Lines 215-216: “These results determine that the total number of perseveration errors may be a variable that shows greater sensitivity to the effects of aging, rather than the number of perseveration errors.” Is there an error here? The Authors are saying that the total number of perseveration errors is more sensitive to age than (again) the number of perseveration errors. What does it mean?

Does IRT have any limits? The Authors are presenting only its advantages but not its limitations. This is a unidirectional way of presenting results, not suitable from a scientific perspective. Could the Authors integrate with a critical reasoning on both pros and cons of IRT?

Other comments

Some formal inaccuracies in drafting the manuscript are still present. For instance, see the following:

  • Line 69. There’s a typo to be addressed: "of the tests with respect to the persons used. of the tests with respect to the persons used"
  • Lines 69 and 71. Moreover, is not appropriate to say the persons or people used. It would be better to say something like “with respect to the person who’s administering the test” or something similar, if this is what the Authors meant.
  • Lines 75 to 77: the same sentence is repeated twice.
  • Line 172: “EA” has not been corrected.
  • Line 192: Theory after IRT is a repetition.
  • Similar issues as in R1 with reference to the use of acronyms: sometime authors use PD and AD, sometimes the extended forms Parkinson’s disease and Alzheimer Disease

Author Response

  1. Thanks for your comment. You are right. We identify the type of study as a cross-sectional study in order to identify the clinical sensitivity of WCST, in addition, we believe as you say that 2.2 Study Population and Clinical Characteristics should be deleted as sub-paragraph, so we join it to study design.
  2. Results
    1. Thanks for this comment, you are right because it could be misunderstood depending on the meaning, we just wanted to highlight results, but you are right due to we used Bonferroni to test the multiple comparisons, this phrase could have eventually been understood as a result that appears to indicate statistical significance in the dependent variable, even when there is none, for that we delate. Thanks so much for your comment. 
    2. Thanks so much for your wonderful question, if we take into consideration age ( 70.71 ± 7.54, 71.81 ± 7.26, 74.75 ± 7.66) of the three groups we have into consideration that the age will be a factor that influences. However, we put it as a confusion bias. Regarding the results, it is necessary at least to say something about age.

Disussion 1 and 2 

We explain it based on our research, we have to take into account that WCST measures the capacity of the evaluated person to face a new conceptual problem and evaluate our ability to generate working hypotheses and to test them, in our research we make that assertion because Demakis (2003) carried out a meta-analysis of 25 studies comparing adults with frontal brain injuries (n = 644) with adults without frontal injuries (n = 705) using the WCST. They concluded that the WCST showed a modest effect of locating frontal lesions. (Cohen's d = –0.37) having a negative relationship it could indicate that there is a lower ability to locate lesions of the frontal cortex.  When the data is analyzed in detail, we observed that an important part of the size of the effect detected with the WSCT could be due to the presence of specific lesions of regions of the dorsolateral prefrontal cortex, taking into account that the investigations carried out in the last decades have revealed that cognitive disorders are part of the clinical symptoms of Parkinson's disease (PD) and point to the frontal lobes as the most affected, we believe that this is why the deviation that we identify in the figure occurs. 

Line 208. OK. You are right we modify for normal functioning. Thanks so much for your comment. 

Lines 215-216:  Thanks for this comment we say it in order to test the non-linear models, basic on the logistic regression, it true, maybe we have over generalized it, so we delate.  Thanks so much for your comment. 

Other comments

Some formal inaccuracies in drafting the manuscript are still present. For instance, see the following:

· Line 69. Thanks so much, we have addressed it 

· Lines 69 and 71. Thanks so much, we have addressed it 

· Lines 75 to 77: Thanks so much, we have addressed it. 

· Line 172: “EA” has not been corrected. Thanks so much, we have addressed it 

· Line 192: Theory after IRT is a repetition. Thanks so much, we have addressed it

In addition, we have homogenise the use of PD and AD

Thanks so much for your review.